# Antimicrobial Peptides from Black Soldier Fly (*Hermetia illucens*) as Potential Antimicrobial Factors Representing an Alternative to Antibiotics in Livestock Farming

**DOI:** 10.3390/ani11071937

**Published:** 2021-06-29

**Authors:** Jing Xia, Chaorong Ge, Huaiying Yao

**Affiliations:** 1Research Center for Environmental Ecology and Engineering, School of Environmental Ecology and Biological Engineering, Wuhan Institute of Technology, Wuhan 430073, China; jingxia51@126.com; 2Zhejiang Key Laboratory of Urban Environmental Processes and Pollution Control, Ningbo Urban Environment Observation and Research Station, Chinese Academy of Sciences, Ningbo 315800, China; 3Key Laboratory of Urban Environment and Health, Institute of Urban Environment, Chinese Academy of Sciences, Xiamen 361021, China

**Keywords:** antimicrobial peptides, *Hermetia illucens*, antibiotic substitute, bacterial resistance, livestock farming

## Abstract

**Simple Summary:**

Microbial resistance to antibiotics is a constant threat to livestock farming, and unreasonable use of antibiotics has increased the prevalence of infectious diseases in humans and animals. Antimicrobial peptides derived from black soldier fly, *Hermetia illucens* (Diptera: Stratiomyidae), have great potential as alternatives to antibiotics for prophylaxis and treatment of diseases in animals because they have extensive antimicrobial properties and a lower tendency to induce resistance. Additionally, several studies have shown that *Hermetia illucens* larvae can participate in a circular economy by digesting organic waste alone and then promoting the growth performance of domestic animals fed the larvae. Therefore, antimicrobial peptides from *Hermetia illucens* are promising candidate for replacement of antibiotics in livestock farming.

**Abstract:**

Functional antimicrobial peptides (AMPs) are an important class of effector molecules of innate host immune defense against pathogen invasion. Inability of microorganisms to develop resistance against the majority of AMPs has made them alternatives to antibiotics, contributing to the development of a new generation of antimicrobials. Due to extensive biodiversity, insects are one of the most abundant sources of novel AMPs. Notably, black soldier fly insect (BSF; *Hermetia illucens* (Diptera: Stratiomyidae)) feeds on decaying substrates and displays a supernormal capacity to survive under adverse conditions in the presence of abundant microorganisms, therefore, BSF is one of the most promising sources for identification of AMPs. However, discovery, functional investigation, and drug development to replace antibiotics with AMPs from *Hermetia illucens* remain in a preliminary stage. In this review, we provide general information on currently verified AMPs of *Hermetia illucens*, describe their potential medical value, discuss the mechanism of their synthesis and interactions, and consider the development of bacterial resistance to AMPs in comparison with antibiotics, aiming to provide a candidate for substitution of antibiotics in livestock farming or, to some extent, for blocking the horizontal transfer of resistance genes in the environment, which is beneficial to human and animal welfare.

## 1. Introduction

Growth of the world population and improvement of living standards in developing countries is linked to growing consumption and demand for animal-derived protein sources and increasing requirements for protein desperately need enhanced livestock production [1]. The golden age of antibiotics began with their discovery. Therefore, livestock farmers irrationally used antibiotics in animal feed to improve animal production performance and to ensure a high yield of animal-derived protein sources [2]. However, the use of antibiotics in animal diets resulted in dissemination of drug resistance among bacteria and decreased quality of meat products [3]. Humans will gradually approach the lack of medicines to cure the infections, and this lack will eventually present a risk to worldwide public health [4]. The use of antibiotics in livestock has been banned in many countries due to adverse consequences [5,6,7,8]. Thus, extensive studies are aiming to identify reliable alternatives to antibiotics [9,10,11].

Recent studies demonstrated that insects are a sustainable source for animal feed in many countries around the world due to their ability to provide nutritional ingredients [12]. Insects have a higher feed conversion efficiency and can exploit the nutrients of the diet better than animals [13]. Moreover, insects produce lower ammonia emission and greenhouse gases than those produced by traditional livestock [14,15]. On the other hand, insects as feed can also indirectly ameliorate the environmental footprint of vertebrate meat production [16]. Feeding insects with human-inedible organic wastes, and then rearing poultry with these insects, facilitates an increase in protein content in these animals [17], which achieves a circular economy [18] of wastes and reduces the consumption of starting material and energy to promote animal growth. This approach will be conducive to overcoming the future paucity of adequate, nutritious, and healthy food in the future.

Black soldier fly (BSF), *Hermetia illucens* (Diptera: Stratiomyidae), is a warm-climate [19] and innocuous insect [20,21] that rapidly colonizes decomposing waste substances, such as food scraps or kitchen waste [22], straw [23,24], manure [25,26]. The variety of substances and efficiency of their consumption by BSF larvae is higher than that by *Drosophila melanogaster*, *Apis mellifera,* and *Bombyx mori* [27]. Habitats of BSF larvae are characterized by abundance of various microorganisms. BSF is able to live in adverse environments, indicating that BSF has an innate immune system that can produce various substances, such as peptides [28], that protect against bacteria, fungi, and viruses [29,30,31]. A defensin-like peptide 4 (DLP4) was recently detected and extracted from various immune tissues of BSF larvae. The survival rate of mice infected with methicillin-resistant *Staphylococcus aureus* (MRSA) treated with DLP4 at the doses ranging from 3 mg/kg to 7.5 mg/kg was 80–100% [32]. This finding demonstrated that DLP4 is a promising novel candidate antimicrobial peptide (AMP) against MRSA infections. AMPs from BSF have been at the epicenter of research for decades and are expected to be able to substitute for antibiotics in poultry feed and, to some extent, block the horizontal transfer of resistance genes in the environment [33].

The present review summarizes the studies on AMPs from *Hermetia illucens* reported in the databases, such as National Center for Biotechnology Information (NCBI, https://www.ncbi.nlm.nih.gov/, accessed on 28 June 2021), antimicrobial peptide database (APD, http://aps.unmc.edu/AP/, accessed on 28 June 2021) and KEGG pathway database (https://www.kegg.jp/kegg-bin/show_pathway?map01503, accessed on 28 June 2021), and provides a comprehensive and structured overview by comparing AMPs from BSF with those from other insect species in the context of six aspects: medicinal value, diversity, mechanism of action, immune-induced signaling pathway, bacterial resistance, and applications in livestock production. Ultimately, relevant information on AMPs from BSF provides an important basis for the development of alternatives to antibiotics to block antibiotic pollution of the environment.

## 2. Medicinal Value of Antimicrobial Peptides

Edible insects have been consumed in China for more than two thousand years. Insects are also a source of various natural substances that can exploit natural bioactive ingredients in medical, veterinary, and agricultural applications [34,35]. Insect toxins are compounds with many biological activities and may be used as drugs for alleviation of pain, treatment of certain diseases and even cancer therapy [36]. Insects are extensively used in conventional medicine worldwide [37]. The number of insect patent applications has rapidly increased since 2010, indicating that the demand and utilization of insect applications continue to increase in China [38].

In Chinese medicine, approximately 300 insect species are used to produce 1700 conventional medications [39,40]. Ants are famous medicinal species, their mandibles are used in surgery to staple wounds and ant-generated substances accelerate wound healing [41]. However, *Drosophila melanogaster* is the most common insect studied over the last few years. For example, the asexual blood stage of *Plasmodium falciparum* parasites causes a series of clinical manifestation of malaria, such as anemia and fever, sometimes resulting in death [42]. Tonk et al. [43] discovered that low concentrations of AMPs (Drosocin, Mtk-1, and Mtk-2) from the fruit fly *Drosophila melanogaster* had low hemolytic effects on mouse and pig erythrocytes but significantly inhibited the growth of *Plasmodium falciparum*. Hence, insect-derived AMPs can be considered candidates for antiparasitic drugs.

The value of insects in medicinal applications is still being explored; for example, BSF larvae before the prepupal stage can be exploited as a high-quality source of protein and oil and a high-content source of chitin, AMPs, and melanin [44,45]. Proteins have been used in the production of aquaculture and poultry feeds because of their high digestibility. Practice indicates that the use of protein derived from the BSF in feed significantly reduces the incidence of diarrhea in breeding animals, improves the growth performance of the animals, and thus promotes the development of breeding history [13,46,47]. High oil content is another valuable property of BSF. The content of oil obtained from *Hermetia illucens* larvae is considerably influenced by feeding materials, however, the content of crude lipid in these larvae is far greater than that in other insects or animal feed sources, such as soybean flour and fish meal [44]. AMPs extracted from *Hermetia illucens* larvae are the principal components of medical value. Li et al. [32] demonstrated that defensin-like peptides 2 and 4 (DLP2 and DLP4) decreased disseminated bacterial burden by over 95% in the spleen and kidneys, reduced serum levels of proinflammatory cytokine, increased the levels of anti-inflammatory cytokine, and repaired lung and spleen injury. This finding suggests that DLP2 and DLP4 extracted from *Hermetia illucens* are promising candidates against staphylococcal infections. Therefore, AMPs from *Hermetia illucens* may be an important subject of investigations in the biomedical field.

## 3. Diversity of Antimicrobial Peptides in Insects

Identified AMPs are very diverse and hard to categorize, and their classification is generally based of secondary structure. Insects account for ninety percent of total quantity of animals on Earth, however, AMPs derived from insects correspond to only approximately ten percent of more than 2830 AMPs listed in the Antimicrobial Peptide Database [32]; thus, additional functional AMPs can be identified in insect species [48]. AMPs with antibacterial activity [49] were initially detected in the hemolymph of giant silk moth pupae, *Samia Cynthia*, in 1974, then, an insect AMP of the cecropin type was purified from the hemolymph of immunized Cecropia moth pupae, *Hyalophora cecropia,* in 1980 [50]. Subsequently, over 150 insect AMPs have been extracted, purified, and identified [48]. Amino acid sequences and structural characteristics of, insect AMPs are used to define four broad categories: (a) α-helical structural peptides (e.g., moricin and ceropin), (b) glycine-rich peptides (e.g., gloverin and attacin), (c) proline-arginine-rich peptides (e.g., apidaecin, metchnikowin, and drosocin), and (d) cysteine-rich peptides (e.g., drosomycin and defensin) [48]. Most active AMPs are composed of 20–50 residues and are synthesized from inactive precursor proteins via limited proteolysis [51,52]. AMPs of the same classes or subclasses display different biological properties in different insect species. For instance, cecropin A, from *Anopheles gambiae*, has both antibacterial and antifungal activities; however, cecropin A from silk moths has only antibacterial activity, and this difference results from differences in the structure and size, which impact the activity of these AMPs toward various microorganisms [53]. Antimicrobial effects may involve enhanced innate immune reactions, and AMPs play a selective immunoregulatory role in infection by participating in wound healing and angiogenesis [54].

Isolation and identification of antimicrobial peptides from *Hermetia illucens* has been progressing in recent years (Table 1) [29,55,56,57,58,59]. Structural similarity or unique sequences of defensin and cecropin AMPs enabled investigation of these AMPs in greater detail compared to other types of AMPs from *Hermetia illucens*. Defensins contain six conserved cysteines and are the most widespread AMPs in insects [60]. Insect defensins commonly include an N-terminal loop, an antiparallel β-sheet and an α-helix, generating a “loop-helix-sheet” or “cysteine-stabilized alpha beta (CSαβ)” structure [61]. Li et al. [32] demonstrated that *Hermetia illucens* defensins DLP2 and DLP4 form a typical CSαβ structure and have a significant antibacterial effects against MRSA, increasing the survival rate of mice challenged with MRSA. Soon-IK et al. [55] studied the cecropin family of AMPs from *Hermetia illucens* and demonstrated that inhibition of *Escherichia. coli* activity by cecropin-like peptide 1 (CLP1), containing the N-terminal helix, was 50-fold greater than that produced by ampicillin. This result demonstrated that N-terminal primary structure of cecropin may be important for antibacterial function. Furthermore, diverse unknown structures and antimicrobial functions of AMPs from *Hermetia illucens* in host defense systems, in almost all living organisms in nature should be explored. Scientists firmly consider that these AMPs to have promising practical applications in the future [58].

## 4. Mechanism of Action of Antimicrobial Peptides

The structures of AMPs are diverse, however, the biological mechanism of the effects of AMPs is related to the destruction of the bacterial cell membrane. The great majority of AMPs contain a cationic structure that binds to anionic lipopolysaccharides (LPS), to teichoic acids, and lipoteichoic acids (Figure 1 part 1). AMPs eventually destroy the integrity of the envelope by forming ion channels or transmembrane pores to trigger leakage of the cell contents that kills the cells [62]. Specific mechanisms of membrane lysis can be categorized into four models [63]: the toroidal pore, carpet-like, barrel-stave, and unstructured ring pore models (Figure 1 part 2). Lipid components of the membrane surface are the primary targets of AMPs. According to the toroidal pore model (Figure 1 part 2a), AMPs bind to the lipid components and form pores, ultimately resulting in disruption of bacterial membrane. According to the second model (Figure 1, part 2b), AMPs enshroud the cell membrane in a carpet-like mode, this mechanism requires a high concentration of AMPs and leads to cell membrane lysis. Cecropin-type AMPs act via this mechanism [63]. For example, Sato et al. [64] demonstrated that continuous accumulation of cecropin-type AMPs at the bacterial lipid bilayer results in the formation of a “carpet-like” peptide structure on the membrane surface. This “carpet-like” structure has inherent detergent-like lytic properties, which dissolve the membranes. The barrel stave model represents the third mechanism (Figure 1, part 2c) of action of AMPs, in this model, the peptides combine with the cell envelope and enter the hydrophobic core of the phospholipid bilayer, resulting in leakage of intracellular substances and a reduction in the membrane potential. AMPs that damage the membranes via this mechanism include such as ceratotoxins [65] and amphotericin B [66], eventually cause cell death. The fourth mechanism (Figure 1 part 2d) involve damage of the cell envelope due to the formation of “unstructured ring pores”; i.e., AMPs line up in a ring-like structure that can be attached to the membrane at various angles [67].

In addition to inducing membrane damage, some AMPs can spontaneously traversing cell membranes, interact with intracellular molecules and thus disrupt intracellular metabolic processes (Figure 1 part 3). Inside the cell, AMPs, such as drosocin and pyrrhocoricin, mainly interact with the bacterial chaperone DnaK (70 kDa) [68], which is likely to inhibit protein folding in the cells and lead to metabolic disorders [69]. Additionally, DNA may be a target. The N-terminal region of AMP lactoferricin has been shown to incorporate into specific DNA domains where it can regulate transcription [70].

The interaction of high concentrations of AMPs with biological membranes induces an antimicrobial effect due to permeabilization or disruption the cell envelope. However, suitable low concentrations of AMPs affect microbial activity or viability due to interactions with intracellular small molecules [68].

## 5. AMP Induction of Immune Signaling Pathways in Insects

Induction of synthesis and secretion of numerous AMPs, lectins, lysozymes, and protein inhibitors upon invasion with exogenous pathogens are the main functions of humoral immunity in insects [71,72,73,74]. AMPs are important response factors of natural humoral immunity that play a role in the initial defense strategy against exogenous invading pathogens. Insects have a wide variety of AMP molecules; however, insect cells mainly produce specific AMPs through the Toll-like signaling and immune deficiency (IMD) pathway (Figure 2). The Toll pathway is largely activated by Gram-positive bacteria, fungi, and virulence factors (such as proteases). During various pathogenic infections, proteins encoded by spaetzle associated genes are recognized by Toll-like receptors on the cell membranes. Binding of spaetzle induces binding of activated Toll-like receptor to the primary response protein MyD88 via cytosolic intracellular homology domain known as Toll/IL-1R (TIR) [75,76,77]. Activation of upstream interactions induces integration of MyD88, interleukin-1 receptor-associated kinase 4 (Tube) and interleukin-1 receptor-associated kinase 1 (Pell) to form a MyD88-Tube-Pelle heterotrimeric complex via death domain (DD)-mediated interactions [77,78,79]. Subsequent, signal results in nuclear translocation of the NF-κB transcription factors Dorsal and/or Dif and the phosphorylation and degradation of the IκB inhibitor Cactus [80]. In the absence of a signal, Cactus binds to the NF-κB transcription factors Dorsal and/or Dif in a context-dependent manner and inhibits their activity and nuclear localization. Therefore, nuclear translocation of both Dorsal and Dif requires Cactus degradation [81]. Finally, the transcription factors Dorsal/Dif translocate into the nucleus and activate the secretion of corresponding AMPs to achieve the defense response [82]. On the other hand, the IMD pathway in insects responds to Gram-negative bacteria and fungi. Peptidoglycan (PGN) monomers or polymers from Gram-negative bacteria and secretions from fungi stimulate PGN recognition protein LC (PGRP-LC) and induce a series of cascade reactions [83]; the N-terminus of the nuclear factor Relish translocates to the nucleus and induces the transcriptional upregulation of AMP expression, which ultimately leads to the formation of AMPs in insects [84]. Additionally, these two signaling pathways act independently and lead to the induction of AMP production by transferring essential proteins into the nucleus [85].

Currently, exact mechanism of production of AMPs in *Hermetia illucens* larvae is unknown. However, Huang et al. [86] demonstrated that treatment of *Hermetia illucens* with *Duox-TLR3* interference RNA deactivates the NF-κB signaling pathway, downregulates AMP expression, and reduces inhibitory effects on zoonotic pathogens. This finding demonstrated that *Hermetia illucens* utilizes the Toll pathway to regulate the expression of AMPs and subsequently inhibit pathogens in adverse environments. Furthermore, the suppression of the activities of Toll-like receptor 2 (TLR2) and 4 (TLR4) considerably diminishes NO production by dipterose-BSF, suggesting that dipterose-BSF provokes the immune function of various cytokines in *Hermetia illucens* macrophages via TLRs [87].

In practice, survival of BSF larvae in environments contaminated with pathogens and conversion of organic wastes into protein- or fat-rich biomass, which can be used as feed substrates for livestock imply that BSF larvae may contain a variety of abundant AMPs that protect the larvae against infection by invading pathogenic bacteria. However, the mechanisms of antimicrobial effects of BSF-derived AMPs are unclear. Therefore, mechanistic investigations of antimicrobial properties of BSF-derived AMPs, which enable the replacement of antibiotics in livestock farming with BSF-derived AMPs are important subjects for future studies.

## 6. Bacterial Resistance to Insect AMPs

Bacterial resistance to antibiotic substances generally depends on drug inactivation and mutations or modifications of the target sites. Reduced accumulation of pharmacologically active substances due to limited uptake (for example, in Gram-negative bacteria) or improved excretion (for example, in Gram-positive bacteria) represents a significant mechanism of resistance to some types of antibacterial agents [88,89]. Another resistance mechanism is based on specific growth modes of bacteria, for instance, biofilm formation [90]. This mechanism easily induces bacterial multidrug resistance and leads to a series of disease outbreaks and large economic losses in stockbreeding [91].

Unlike common antibiotics, such as kanamycin, ampicillin, and ciprofloxacin, which trigger a three- to fourfold augmentation of bacterial mutation rates, cationic antimicrobial peptides do not enhance the mutation rate in bacteria [92]. Moreover, Rodríguez-Rojas et al. demonstrated that stress-mediated channels enhanced the mutation rates of *Escherichia coli* only upon exposure to common antibiotics. In contrast, AMPs (cecropin A and melittin) extracted from insects did not induce bacterial stress pathways [92]. Therefore, these findings provide a new perspective suggesting that AMPs provides a distinct advantage to prevent the development of drug resistance because AMPs do not stimulate adaptation of bacteria to these immune defenses.

Horizontal transfer of bacterial resistance genes is a common and distinctive pattern of acquisition of antibiotic resistance acquisition. This process involves consolidation of drug resistance genes into DNA to create various clusters responsible for resistances of bacterial species in the environment. The mechanism of bacterial resistance to AMPs from BSF has not been described in detail. However, the development of bacterial resistance may involve inactivation of antibiotics by structural modification, alterations in the targets of antibiotics, and rapid removal of antibiotics from bacterial cells by an efflux pump-mediated mechanism [93]. This explanation for the development of bacterial drug resistance may be associated with AMPs entering the cells. Changes in the cell envelope are one of the main mechanisms of bacterial resistance against AMPs [92]. In the case of Gram-positive bacteria, relevant studies demonstrated that an increase in the levels of D-alanine esters in teichoic acids alters the charge on the surface of the cell wall, which facilitates the development of resistance of *Staphylococcus aureus* to vancomycin and defensins [54,94]. Gram-negative bacteria can avoid the effects of defensins via modification of their external cell envelope by acylating lipid A in the lipopolysaccharide layer to induce AMP resistance [95]. On the other hand, the production of a polysaccharide capsule is a mechanism of resistance of *Klebsiella pneumoniae* against defensins [54,96]. Enzymatic antibiotic inactivation is another common mechanism of resistance, which involves enzymes produced by resistant bacteria. This resistance mechanism includes modifications of a various antibiotic molecules by transfer of functional groups, such as acetyl, phosphoryl, ADP-ribosyl and glycosyl moieties [97]. The most representative enzymes include β-lactamases, which hydrolyze the C-N bond and thus decrease antibiotic activity [93]. Unfortunately, only a few studies investigated the degradation of insect AMPs by bacterial enzymes. The results indicated that *Staphylococcus aureus* resists human α-defensins by generating staphylokinase, which neutralizes bactericidal effects of these AMPs [98]. Additionally, proteases degrading AMPs which lack terminal charged residues, have been identified in *Escherichia coli* [88] and *Salmonella enterica serovar Typhimurium* [99]. On the other hand, antimicrobial peptides can manifest lytic effects on eukaryotic cells, hence, intracellular proteins may be released due to peptide-dependent lysis [100]. Thus, these factors may limit the use of AMPs.

Recently, natural composites containing multiple insect AMPs were shown to induce bacterial resistance at a considerably lower rate than that of individual peptides and small-molecule antibiotics [101]. The authors extracted AMP complexes (defensins, cecropins, and diptericins), which belong to the families of cytomembrane-disrupting/permeabilizing peptides, from the blow fly *Calliphora vicina* infected with a mixture of *Escherichia coli* D31 and *Micrococcus luteus* A270. Moreover, the authors analyzed the changes in the resistance using *Escherichia coli, Klebsiella pneumoniae,* and *Acinetobacter baumannii* strains under selective stress of AMPs compound and control groups of antibiotics. All tested bacteria easily acquired drug resistance to antibiotics; in contrast, indications of the development of resistance to the AMP complex were not detected. This finding demonstrated that natural AMP complexes may provide novel solutions to the drug resistance problem. Similarly, BSF can be stimulated to secrete many kinds of AMPs in response to hostile environments containing a multitude of microbial species [102]. Similarly, Antonio et al. [103] identified 57 putatively active peptides in BSF by machine learning bioinformatics algorithms.

The risk of the development of bacterial resistance against AMPs from BSF is considered relatively low, however, subsequent experimental validation of this hypothesis is needed with regard to potential risk of applications of these peptides.

## 7. Application of BSF-Derived AMPs in Livestock Production

BSF has been investigated for the ability to convert organic waste material to high-quality protein, inhibit certain detrimental bacteria and vermin, and furnish possible chemical precursors for synthesis of biodiesel and for possible use as a feed source for diverse animals [104]. The body composition of BSF larvae is determined by quality and quantity of feed intake [105]. For example, BSF larvae fed pig manure have higher protein content than those fed cow dung [106,107,108]; however, a diet based on grain waste leads to an even higher protein level [109]. The same is true for crude fat levels. Nguyen et al. [105] demonstrated that BSF larvae reared on liver and fish had higher fat and protein content than those reared on chicken meal. Moreover, the body composition undergoes considerable changes during larval development. For instance, crude protein content has been shown to decrease during the progress through the larval instar stage, namely, the crude protein content (dry matter) of five-day-old larvae was the highest (61%) and that of 15-day-old (44%) and 20-day-old (42%) larvae was the lowest [110]. The changes in the amino acid content in dried BSF larvae were also associated with the larval diet. For example, the amino acid content of BSF larvae fed cattle manure tended to be slightly higher than that of the larvae fed pig dung or chicken dung [107,111]. Apparently, the lysine content (6–8%) was exceedingly high in BSF larvae compared with that in characterized animal feeds [112]. For example, the levels of some essential amino acids in BSF larvae reared on pig slurry are consistent with those in soybean powder, especially the levels of leucine, threonine, phenylalanine and lysine [108]. Moreover, comparison with soybean meal indicated that BSF larvae contain higher levels of tryptophan, alanine, methionine, and histidine and lower levels of arginine. BSF larvae have a high protein content and contain many essential nutrient ingredients in the vast majority of animal diet compositions, which can impact the taste or digestibility of BSF larval meal [113].

Currently, the health and productivity benefits of poultry are major concerns in the poultry feeding industry. Pathogenic microorganisms, such as bacteria, viruses, and parasites, remain important factors affecting livestock health [114]. For example, viral or bacterial diarrhea in pigs [114], mastitis in dairy cows caused by *Staphylococcus aureus* [115] and coccidiosis in chickens infected with *Eimeria tenella* [116] can cause economic losses for farmers. Therefore, the growth performance of livestock poultry is directly influenced by feeding management, feeding environment, and diseases. Furthermore, AMPs whose production is induced in BSF larvae have certain advantageous characteristics, such as low molecular weight, high thermal stability, broad antimicrobial spectrum, and specific mechanism of action. In particular, a significant antimicrobial effect on drug-resistant bacteria has been demonstrated. Certain immune recognition function of BSF AMPs have also been shown these AMPs do not act on normal eukaryotic cells and only act on prokaryotic cells and pathological eukaryotic cells, and these properties can reduce the side effects of clinical treatment (such as an increase in hemoglobin (HGB) and hematocrit (HCT) in the blood, which contribute to enhanced oxygen-binding capacity and transport of the oxygen to the tissues of the body) [117]. A study demonstrated the lack of adverse effects on growth performance, serum indexes, diarrhea rate or nutrient digestibility in weaned piglets fed BSF larva meal when prepupa meal was used to replace an the appropriate proportion of soybean meal, fish meal, and plasma protein meal [118]. Furthermore, nutrient digestibility, growth performance, and immune capacity of weaned piglets were improved when BSF larvae fed kitchen waste were used to replace fish meal and soybean meal [119]. Thus, BSF meal will be a promising feed source.

The footprint of BSF in the entire ecosystem, is characterized by the nutrient recycling ability of BSF. On the other hand, Insects emit less ammonia and greenhouse gases than other animals [15]. Additionally, rearing of BSF larvae requires less land and less water than rearing of other animals [14]. Moreover, Liu et al. [120,121] demonstrated that BSF larvae effectively degrade the parent compound oxytetracycline (OTC) due to metabolic ability and function of intestinal microorganisms of BSF larvae. Moreover, OTC was not detected in the larval tissue and did not significantly influences the BSF larvae themselves because of their biological properties. Thus, OTC degradation by BSF larvae is an economical and practical means to decrease or remove antibiotic residues in the environment.

To ensure the yield of animals in the livestock industry, farmers add antibiotics to the feed to reduce the incidence rate of animal diseases [122]. However, these actions result in the presence of large quantities of antibiotic residues in the environment, which destroy the original ecological balance and pose a threat to public health [123,124]. Shin et al. determined the cDNA sequence of a BSF-derived AMP (attacin-like) by rapid amplification of cDNA ends-polymerase chain reaction (RACE-PCR) and DNA sequence analysis [58]. Moreover, the BSF-derived AMP gene (attacin) was expressed in the form of an inclusion body in a prokaryotic host, which contributed to reduced toxicity of this AMP to prokaryotic hosts and stabilized the expression of AMPs, demonstrating favorable antimicrobial activity against Gram-positive and Gram-negative bacteria. AMPs with a simple structures are considered to be non-absorbable in the gut and are delivered in the bacterial targets, which have sufficient affinity [125]. AMPs from insects are produced by ribosomes and consist of natural amino acids, hence, their activity in the animal digestive system has no systemic effects on the animals [88]. Moreover, cationic properties of AMPs preferentially influence negatively charged cells, such as microorganisms or cancer cells. However, the impact of AMPs on positively charged eukaryotic cells is limited [126]. In contrast to antibiotics, BSF-derived AMPs may provide a defensive effect that protects the animals from infections by pathogenic microorganisms and alter cellular behavior in response to external damage [127]. Ultimately, application of BSF-derived AMP in livestock production may be considered safe [121].

The practical application of AMPs from BSF is also hampered by the cost of peptide purification and production, susceptibility of AMPs to proteolytic degradation, which has been reviewed previously [100], and allergic reactions of the animals to AMPs [128]. Leni et al. [128] identified two immunoreactive protein fragments in BSF protein hydrolysate, and suggested that tropomyosin is a potential allergen. However, they also confirmed that enzymatic hydrolysis is an effective strategy to reduce allergenic risk of BSF [128]. Accumulation of heavy metals (such as cadmium) and in the ecological restoration of BSF also limit its use in animal feed production [129]. However, Lalander et al. [130] and Gao et al. [131] assessed the accumulation of five types of antibiotics, including roxithromycin, trimethoprim, sulfamethazine, sulfamethoxazole, and sulfamonomethoxine (0.1–10 mg/kg). The results indicated the lack of accumulation of these antibiotics in BSF larvae. Conversely, sulfadiazine accumulated in BSF larvae when antibiotic concentrations ranged from 1 to 10 mg/kg. These studies indicated that antibiotics accumulate in BSF larvae upon treatment of organic substrates with high levels of these compounds. Thus, potential barriers, such as bioaccumulation of medical drugs, heavy metals, and natural toxins, can be controlled and addressed in mass rearing setups through quality control of their rearing substrates.

## 8. Conclusions and Prospects

BSF larvae have broad potential applications for the development and can be used as a resource due to rapid reproduction, large biomass, extensive feeding, harmlessness of adults, and high absorption and conversion rates. At the same time, BSF is a saprophytic resource insect that can feed on livestock excrement and household garbage to produce high-value animal protein feed. The BSF represents a new treatment mode for waste resource utilization with low energy consumption and high output value, which is conducive to the improvement of ecological environment. Moreover, some studies reported good antimicrobial properties of BSF-derived AMPs. The mechanisms of action of AMPs in insects have evolved over hundreds of years, and AMPs are conserved, indicating that the risk for the development of bacterial resistance to BSF-derived AMPs may be low. Hence, BSF-derived AMPs may be promising alternatives to antibiotics in the livestock industry required due to a global problem of increasing bacterial resistance to antibiotics. Therefore, recent, applications of BSF-derived AMPs have become an important subject in biology, agricultural science, medicine science, food, and feed industries. However, exploration of AMPs from BSF is in early stages, and investigations of the mechanisms by which BSF-derived AMPs inhibit pathogenic bacteria and interact with the resistance genes are lacking, although some studies on AMP extraction from *Hermetia illucens* and antibacterial activities have been carried out.

Abuse of antibiotics in various areas results in unsolved scientific questions about drug resistance in the environment caused by antibiotic contamination. Recent, detection and identification of AMPs from BSF were accelerated due to extraordinary superiority of these agents. Therefore, potential use of these AMPs as alternatives to antibiotics requires in-depth research. Future, studies on BSF AMPs and their modes of action should explore the following aspects: (a) validation of AMPs from BSF using various immune induction approaches; (b) identification of AMPs by pathogen recognition receptors and downstream reaction cascades leading to the production of BSF AMPs; (c) studies of the molecular mechanism of action of BSF AMPs against pathogenic microorganisms; (d) characterization of interactions between *Hermetia illucens* AMPs and resistance genes in antibiotic-resistant microorganisms; and (e) evaluation and application of BSF AMPs as alternatives to antibiotics.

## Figures and Tables

**Figure 1 animals-11-01937-f001:**
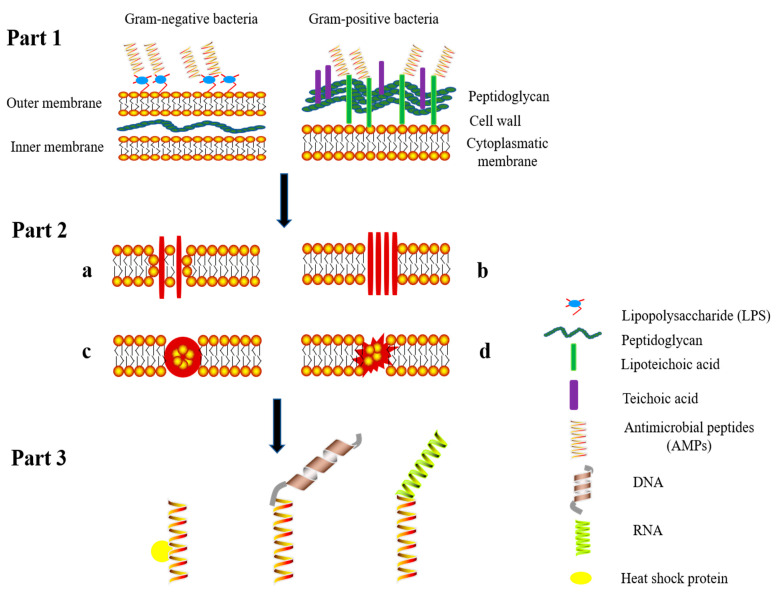
Mechanism of action of antimicrobial peptides on bacterial cells. This figure presents a scheme of the models of action of AMPs: binding to the bacterial cell membrane (Part 1), possible effect resulting in the destruction of bacterial cell membrane (Part 2) and interactions of AMPs with intracellular substances (Part 3). Part 1: AMPs bind to lipopolysaccharides (LPS) of Gram-negative bacteria and to lipoteichoic or teichoic acid of Gram-positive bacteria and penetrate the cell wall. Part 2: Then the AMPs destroy the membrane structure via four pathways (**a**) toroidal model, (**b**) carpet-like model, (**c**) barrel-stave model, and (**d**) unstructured ring pores. Part 3: antibacterial activity of AMPs is mediated by interactions with heat shock proteins, DNA and RNA.

**Figure 2 animals-11-01937-f002:**
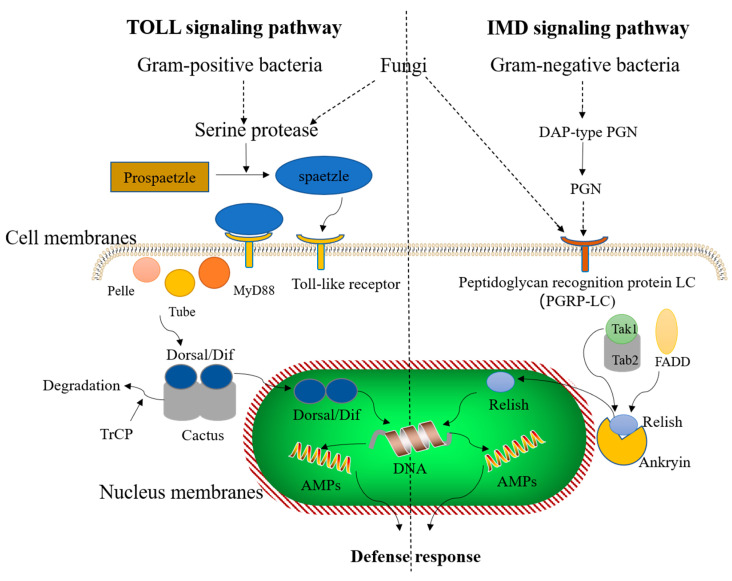
The immune-induced signaling pathway of insect antimicrobial peptides (AMPs). The Toll signaling pathway is activated by Gram-positive bacteria and fungi, and the immune deficiency (IMD) signaling pathway is activated by Gram-negative bacteria and fungi. These two signaling pathways act independently and AMP production is induced by transport of a series of necessary proteins into the nucleus.

**Table 1 animals-11-01937-t001:** Antimicrobial peptides from *Hermetia illucens*.

Peptide	Amino Acid Sequence	Immune-Induced Strains	Reference
Defensin			
Defensin-like peptide 1 (DLP1)	MRSVLVLGLIVAAFAVYTSAQPYQLQYEEDGLDQAVELPIEEEQLPSQVVEQHYRAKRATCDLLSPFKVGHAACALHCIALGRRGGWCDGRAVCNCRR	*Staphylococcus aureus* KCCM 40881	[59]
Defensin-like peptide 2 (DLP2)	MRSILVLGLIVAAFAVYTSAQPYQLQYEEDGPGYALELPSEEEGLPSQVVEQHYRAKRATCDLLSPFKVGHAACALHCIAMGRRGGWCDGRAVCNCRR	*Staphylococcus aureus* KCCM 40881	[59]
Defensin-like peptide 3 (DLP3)	MRSILVLGLIVAVFGVYTSAQPYQLQYEEDGPEYALVLPIEEEELPSQVVEQHYRAKRATCDLLSPFGVGHAACAVHCIAMGRRGGWCDDRAVCNCRR	*Staphylococcus aureus* KCCM 40881	[56,59]
Defensin-like peptide 4 (DLP4)	MVHCQPFQLETEGDQQLEPVVAEVDDVVDLVAIPEHTREKRATCDLLSPFKVGHAACAAHCIARGKRGGWCDKRAVCNCRK	*Staphylococcus aureus* KCCM 40881	[59]
Defensin 1 (HiDef1)	unknown	*Lactobacillus casei*	[57]
Cecropin			
CecropinZ1	GWLKKIGKMKFILGTTLAIVIAIFGQCQAATWSYNPNGGATVTWTANVAATAR	*Escherichia. coli* and *Staphylococcus aureus*	[29]
Cecropin 1 (Hicec1)	unknown	*Lactobacillus casei*	[57]
Cecropin-like peptide 1 (CLP1)	MNFTKLFVVFAVVLVAFAGQSEAGWRKRVFKPVEKFGQRVRDAGVQGIAIAQQGANVLATARGGPPQQG	*Staphylococcus aureus* KCCM 40881	[55]
Cecropin-like peptide 2 (CLP2)	MNFAKLFVVFAIVLVAFSGQSEAGWWKRVFKPVEKLGQRVRDAGIQGLEIAQQGANVLATARGGPPQQG	*Staphylococcus aureus* KCCM 40881	[55]
Cecropin-like peptide 3 (CLP3)	MNFTKLFVVFAVVLIAFSGQSEAGWWKRVFKPVERLGQRVRDAGIQGLEIAQQGANVLATVRGGPPQQG	*Staphylococcus aureus* KCCM 40881	[55]
Attacin			
*Hermetia illucens*-attacin	MASKFLGNPNHNIGGGVFAAGNTRSNTPSLGAFGTLNLKDHSLGVSHTITPGVSDTFSQNARLNILKTPDHRVDANVFNSHTRLNNGFAFDKRGGSLDYTHRAGHGLSLGASHIPKFGTTAELTGKANLWRSPSGLSTFDLTGSASRTFGGPMAGRNNFGAGLGFSHRF	*Escherichia. coli* KCCM 11234	[58]
Sarcotoxin			
Sarcotoxin1	GWLKRKIGMKFILGTTLAIVVAIFGQCQAATWSYNPNGGATVTWTANVAATAR	*Escherichia. coli* and *Staphylococcus aureus*	[29]
Sarcotoxin (2a)	GWLKRKIGKKFILGTTLAIVVAIFGQCQAATWSYNPNGGATVTWTANVAATAR	*Escherichia. coli* and *Staphylococcus aureus*	[29]
Sarcotoxin (2b)	GWLKRKIGKKFILGTTLAIAVAIFGQCQAATWSYNPNGGATVTWTANVAATAR	*Escherichia. coli* and *Staphylococcus aureus*	[29]
Sarcotoxin3	GWLKRKIGMMMKNSNFNSTEEREAAKKNYKRKYVPWFSGANVAATAR	*Escherichia. coli* and *Staphylococcus aureus*	[29]
Stomoxyn			
StomoxynZH1 (a)	RGFRKHFNNLPICVEGLAGDIGSILLGVGSDIGALAGAIANLALIAGECAAQGEAGAAVVAAT	*Escherichia. coli* and *Staphylococcus aureus*	[29]

## Data Availability

Data access can be requested on demand from the corresponding author.

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
