# Peer review of "Antimicrobial Peptides from Black Soldier Fly (Hermetia illucens) as Potential Antimicrobial Factors Representing an Alternative to Antibiotics in Livestock Farming"

_animals, 2021, doi:10.3390/ani11071937_

Round 1
Reviewer 1 Report
The manuscript quality has been significantly improved form the first submitted version. In general, I suggest to pay attention to the punctuation and spacing, since sometimes there are double or no spaces between words or the word and the reference. In order to improve the manuscript, I have only few comments that need to be addressed:
Line 47: I suggest using the past tense, since today the irrational use of antibiotics is banned.
Line 76: Correct the punctuation.
Line 83: “AMP with antimicrobial activity” is a repetition, since AMP stays for antimicrobial peptides.
Line 86: Please, provide the extensive form for KEGG
Line 109: Please correct “et al.” with the correct punctuation.
Line 118: Correct the typo.
Line 125: Please add the reference after Li et al.
Line 167: Correct the typo.
Line 218: Correct et al.
Line 366: List the reference after the author’s name.
Line 385: Add a space between the parenthesis and “and”.
Line 454: Correct the “et al.” form and provide the reference.
Line 462: Remove the bold
Author Response
please see attachment "Response to Reviewer 1 Comments‘’

Reviewer 2 Report
thank the authors for taking my review into account. I have read the manuscript again and the authors have improved the readability of the work. In my opinion, the article is suitable for publication.
Author Response
please see attachment "Response to Reviewer 2 Comments‘’

This manuscript is a resubmission of an earlier submission. The following is a list of the peer review reports and author responses from that submission.
Round 1
Reviewer 1 Report
Authors prepared a very interesting review on antimicrobial peptides of black soldier fly as alternatives to antibiotics in livestock. I really appreciated the work that have been done and the innovative argument related to insects and their derived compounds introduction in animal feeding.
There are some aspects that have to be improved before considering the manuscript for the publication as detailed below.
Simple summary:
Line 16: the expressions “arbitrary employ” and “regulatory injunctions” are not clear. Please, revise the sentence.
Line 18: The use of adjective “tremendous” is probably too much speculative.
Line 22-23: I clearly agree with the important potential related to the AMP of black soldier fly but this sentence is too much speculative since a real alternative to antibiotics still do not exist. I suggest modulating the world “perfect”.
Abstract:
Line 25: What do you mean with “invading pathogens”? Are you sure that microorganisms are completely unable to develop resistance against insects’ AMP?
Line 28: Revise the English
Line 30: What do you mean with “decaying materials”.
Line 30-31: Is there an evidence that AMP are produced due to the presence of microorganism in the feeding biomass of black soldier fly?
Introduction:
Line 43-44: Please, substitute “subsistence level” with more appropriate expression.
Line 49: Antibiotics as growth promoters have been involved in almost all animal species, not only poultry.
Line 51: “cure such maladies”??
Line 53: substitute “growth accelerators” with growth promoters.
Line 54: The limitation related to antibiotic use in livestock was also introduced in Europe. Please, add it in the list.
Line 55: Revise the English.
Line 60-61: This sentence is not clear. Do you referring that insects have a higher feed efficiency compared to animals exploiting better the nutrients of the diet?
Line 65: Revise the English.
Line 70: What do you mean as “rapidly parasitic in decomposed”
Line 76: I suggest you clarify that insects does not have an adaptive immune system but they only count on a powerful innate one (Rosales et al., 2017).
Line 87: Which database?
Medicinal value of antimicrobial peptides
Line 102: Please, add a reference
Line 104: Please, add a reference
Line 111: More information related to the medical relevance of Plasmodium falciparum should be provided.
Line 117: Please, provide a reference.
Line 123: Provide a reference
Line 126: Provide the extended form for all acronyms as the first time they appear in the text (check them in all the manuscript)
Line 128: correct “antiinflammatory”
Line 130: Please, modulate the sentence since “outstanding” could be too much speculative.
Line 133: Why did you assert that chitin may be a hotspot in the biomedical field since you focused your research only on AMP?
Line 141: Please provide the Antimicrobial Peptide Database reference
Line 169-170: Add the reference after the author’s name Soon-IK et al.
Table 1: Avoid the use of interrogative mark (?) substituting it with “unknown” if not available.
Mechanism of action of antimicrobial peptides
Line 205: Please, substitute “demolition” with a more appropriate word.
Line 212: Provide a reference for the four cited models of action.
Line 217: It is not enough to assert “Cecropin-type AMPs act in this way [61]”. Please complete this statement providing more information
Line 228: Provide more information related to DnaK
AMP induction of immune signalling pathways in insects
Line 256-262: The mechanisms that lead to production of AMP is not properly clear. I suggest revising this part in order to better clarify this complex pathway.
Line 276: Add the reference after Huang et al.
Bacterial resistance to insect AMPs
This paragraph is one of the most interesting part of your review. However, it should be better organized since some information are not properly organized appearing separated (e.g., antimicrobial resistance mechanism). I suggest you introduce the antibiotic resistance mechanisms together and then report literature studies that did not reveal a resistance development to AMP. Another aspect should be clarified: could AMP lead to induce antibiotic resistance in bacterial cell (Line 317-318)?
At some point there are controversial information, since you assert that AMP could not be degraded from bacteria and later you reported that E. coli and Salmonella produce proteases that can degrade AMP. In light of this, try to provide a clear vision with potentialities and limits of AMP use.
Line 299: “intractable problems in stock breeding”
Line 306: add a reference
Application of BSF-derived AMPs in livestock and poultry production
Why did you decide to distinguish between livestock and poultry production?
Line 363: Provide the reference after the author’s name.
Line 368: Did you report the protein values expressed as percentage of dry matter or fresh weight?
Line 371: Substitute “cow dung” with “cattle manure”
Line 374: Pig slurry
Line 377: add a reference.
Line 379: What do you mean with “zoon diets”?
Line 380: Why did you separate livestock from poultry?
Line 388: Why the molecular weight should be an advantage?
Line 389: You described before a wide range of mechanisms of action. Why did you report “unique” in this sentence?
Line 394: How they can reduce the side effects of clinical treatments?
Line 397: Add a reference
Line 399-400: Revise the English
Line 406: I think that this could be extended to all animal species not only compared to cattle.
Line 415: Substitute “stockbreeding” with more appropriate term
Line 421-423: This sentence should be better clarified. Did the authors express AMP in a bacterial cell? Was it toxic for the host?
Conclusion and prospects
Line 446: Why only for the poultry industry?
Authors Contributions: Please provide the authors contributions following the CREdiT guidelines as requested form Animals journal.
Reviewer 2 Report
In this review, the authors tried to gather the general knowledge regarding the currently verified AMPs of Hermetia illucens, explore their potential medical value, discuss their mechanism of synthesis and interaction, and consider the development of bacterial resistance.
Given this report and the literature, I believe this to be a interesting manuscript and important to the readers in the field of animal husbandry, entomology, feed science, edible insects and pharmacology. Based on my experience i is proper and up-to-date, because study on edible insect especially on BSF are developing very rapidly.
The content of the article is correct, although in some cases it may mislead the reader.
However, due to the reviewer's duty, I must point out the following problems:
Major flaws:
The most important problem with this article is the English language. Authors often use heavy sentence syntax, incorrect terms, expressive words that significantly affect the scientific quality of the article. I am asking the authors to stick to the scientific language typical of articles. In the appendix to the review, I only highlighted some of the errors. This can significantly affect the perception of readers. The low quality is especially evident at the beginning of the article. Before possible publication, authors must devote a lot of time to make the article clear and fully understandable for the reader. Please check the meanings of some words carefully. We cannot say, for example, that "BSF parasites" on food. The authors and editor should keep in mind that the language of the article could also influence the review of some information.
The authors in their work overestimate the use of antibiotics in animal husbandry. The authors rely on individual articles from one region of the world. After reviewing this article, as a reader, I would have the impression that antibiotics are constantly being used in every step of animal husbandry, regardless of the country. To my knowledge, this is certainly not the case. I am asking for a more broader approach to this topic.
Finally, what I miss the most in this article are moderation and criticism. Article focuses too much on the positive, leaving no room for potential negative effects. what about the allergic nature of these proteins? Also, Hermetia illucens has the feature of accumulating heavy metals (which is why it is used by entomoremediation). Speaking of antibiotics, since the authors say that their use is so frequent and widespread, BSF, living on the faeces of animals (some antibiotics and metabolites are excreted with them) will not accumulate some antibiotics? The possibility of using AMPs is undeniable, but when discussing this topic, it is also necessary to focus on the aspects that may have a negative effect on the animal's organism.
As mentioned, in the attachment I am sending an article with suggestions that require reflection.
In my opinion, the article might published, due to the innovative approach to the subject of edible insects. However, the authors must focus primarily on improving the language and readability of the article. They must also pay attention to the misuse of certain phrases. After extensive revision of the entire article, it should be reviewed again.
